# Mitochonic Acid 5 Improves Duchenne Muscular Dystrophy and Parkinson’s Disease Model of *Caenorhabditis elegans*

**DOI:** 10.3390/ijms23179572

**Published:** 2022-08-24

**Authors:** Xintong Wu, Satoi Nagasawa, Kasumi Muto, Maiko Ueda, Chitose Suzuki, Takaaki Abe, Atsushi Higashitani

**Affiliations:** 1Graduate School of Life Sciences, Tohoku University, Sendai 980-8577, Japan; 2Biomedical Research Core, Tohoku University Graduate School of Medicine, Sendai 980-0872, Japan; 3Department of Clinical Biology and Hormonal Regulation, Tohoku University Graduate School of Medicine, Sendai 980-0872, Japan

**Keywords:** MA-5, mitochondrial calcium, mitochondrial fragmentation, muscular dystrophy, Parkinson’s disease, rotenone

## Abstract

Mitochonic Acid 5 (MA-5) enhances mitochondrial ATP production, restores fibroblasts from mitochondrial disease patients and extends the lifespan of the disease model “Mitomouse”. Additionally, MA-5 interacts with mitofilin and modulates the mitochondrial inner membrane organizing system (MINOS) in mammalian cultured cells. Here, we used the nematode *Caenorhabditis elegans* to investigate whether MA-5 improves the Duchenne muscular dystrophy (DMD) model. Firstly, we confirmed the efficient penetration of MA-5 in the mitochondria of *C. elegans.* MA-5 also alleviated symptoms such as movement decline, muscular tone, mitochondrial fragmentation and Ca^2+^ accumulation of the DMD model. To assess the effect of MA-5 on mitochondria perturbation, we employed a low concentration of rotenone with or without MA-5. MA-5 significantly suppressed rotenone-induced mitochondria reactive oxygen species (ROS) increase, mitochondrial network fragmentation and nuclear destruction in body wall muscles as well as endogenous ATP levels decline. In addition, MA-5 suppressed rotenone-induced degeneration of dopaminergic cephalic (CEP) neurons seen in the Parkinson’s disease (PD) model. Furthermore, the application of MA-5 reduced mitochondrial swelling due to the *immt-1* null mutation. These results indicate that MA-5 has broad mitochondrial homing and MINOS stabilizing activity in metazoans and may be a therapeutic agent for these by ameliorating mitochondrial dysfunction in DMD and PD.

## 1. Introduction

We have developed a mitochondrial homing drug, Mitochonic Acid 5 (MA-5: 4-(2,4-difluorophenyl)-2-(1H-indol-3-yl)-4-oxobutanoic acid), which was synthesized as a derivative of the plant hormone, indole-3-acetic acid (IAA) [1]. Interestingly, IAA is also found in human organs at micromolar concentrations and accumulates in patients with renal failure [2,3,4]. MA-5 was screened as a derivative that increases cellular ATP levels [1]. It also reduces mitochondrial reactive oxygen species (ROS) production, protects cells with mitochondrial dysfunction from fibroblast death and prolongs the survival of a mitochondrial disease model mouse “Mitomouse”, that contains a mtDNA deletion mutation [1,5,6,7].

The most severe and common muscular dystrophy, Duchenne muscular dystrophy (DMD), is a serious progressive muscle disease caused by mutations in the DMD (dystrophin-encoding) gene. In the meantime, Parkinson’s disease (PD) is the second most prevalent neurodegenerative disease in the world. There is increasing evidence that mitochondrial dysfunction leads to progressive deterioration in patients with DMD or PD [8,9]. The underlying pathophysiologies of DMD and PD are complex, but mitochondrial dysfunction is a prominent early consequence established in both [10,11]. At present, treatments of these diseases are largely targeted at controlling the symptoms and focus on maximizing quality of life. Current standard pharmaceutical treatments such as the corticosteroid prednisone for DMD, and L-DOPA and dopamine agonists for PD are also associated with several undesirable side effects [12,13]. Therefore, although there are still no satisfactory therapies available for mitochondrial disorders, pharmaceutical treatments that enhance mitochondrial function or treat the consequences of mitochondrial dysfunction are considered to be effective not only for mitochondrial diseases but also for DMD and PD. Therefore, the novel MA-5, which is effective against mitochondrial disease models [1,5,6,7], is also expected to be effective against skeletal muscle dysfunction in DMD and dopaminergic neurodegeneration in PD.

The nematode *Caenorhabditis elegans* (*C. elegans*) provides the advantage of conducting aging and disease model studies due to the homology it shares with human genome [14]. *C. elegans* has approximately 20,000 protein-coding genes, about the same number as humans, and most proteins involved in basic cell function and metabolism are mammalian homologues [15]. In particular, many genes associated with human diseases such as *dys-1* (an ortholog of human DMD) and *pdr-1* (an ortholog of human PRKN) are highly conserved in nematodes at the molecular level [16,17]. Furthermore, green fluorescent protein (GFP)-based live-imaging techniques can visualize neuromuscular dysfunctions in *C. elegans*. As an example, in studying PD, the administration of the neurotoxins 6-hydroxydopamine (6-OHDA) leads to the visualization of dopaminergic neurodegeneration in the *dat-1p::GFP* transgenic *C. elegans* TG2435 [18,19]. Similarly, chronic exposure to low concentration (2–4 μM) of rotenone, an inhibitor of mitochondrial electron transport chain (ETC) complex I, causes dopaminergic neurodegeneration not only in rodents but also in *C. elegans* [19,20,21,22,23,24]. Moreover, in *C. elegans* upon overexpression of human mutant α-synuclein, similar dopaminergic neurodegeneration can be visualized in an age-dependent manner [25]. Recently, using Cas-CRISPR technology, if there is in sufficient conservation in the *C. elegans* homologue, the target protein can be replaced by one encoded by the human gene [26,27]. These disease and aging models have been used to develop novel drugs [15,28,29,30,31,32]. In this study, we evaluated whether MA-5 could alleviate the manifestations in body wall muscle (BWM) cells in a *C. elegans* DMD model and alleviate the changes in dopaminergic neurons in a *C. elegans* PD model.

## 2. Results

### 2.1. Penetration and Homing Activity of MA-5 into Intact C. elegans Mitochondria

In normal human fibroblasts, fluorescence-labeled MA-5 (BODIPY-MA-5) efficiently penetrates mitochondria as mitochondrial homing activity [5]. To confirm the penetration of MA-5 into *C. elegans*, wild-type adult hermaphrodites were treated with BODIPY-MA-5 solution for 30 min. The green fluorescent signals of BODIPY-MA-5 were observed on mitochondria in BWM cells, germ cells and intestine and closely matched the blue fluorescent signals of AIE™ Mitochondria Blue (Figure 1A,B). In contrast, BODIPY on its own effectively stained intestinal lipid droplets and germ cell membranes [33] (Figure 1A). These clearly show that MA-5 can penetrate almost all mitochondria in intact living *C. elegans.*

Administration of MA-5 at a final concentration of at least 20 μM even increased the median lifespan of ATU3301 animals by about 1 day, but the change was not significant (Table 1). Subsequent experiments were treated at a final concentration of 10 μM, as used in human patient fibroblasts [1].

### 2.2. Alleviation of C. elegans DMD Model Symptom by MA-5

Similar to the human DMD, the *C. elegans dys-1* (*eg33*) mutation synthesizes a C-terminal truncated dystrophin protein that loses scaffolding function [35]. In the *dys-1* (*eg33*) mutant adults synchronized on day 2, a decrease in motility (thrashing rate) was significantly attenuated by treatment with MA-5 (Figure 2A). In addition, the movement of wild-type (WT) worms was slightly but significantly increased by MA-5 treatment. Next, using the *goeIs* GCaMP sensor, we studied muscular cytoplasmic Ca^2+^ ([Ca^2+^]_cyto_) cycling with contraction and relaxation. In the BWM cells of a WT worm immobilized with microspheres, the full-width half-maximum (FWHM) time of delta [Ca^2+^]_cyto_ was around 10 s for each cycle (Figure 2B,C, Appendix A). On the other hand, the FWHM of *dys-1* mutants was broadened to more than 20 s, indicating that the *dys-1* mutant had a longer accumulation of [Ca^2+^]_cyto_, similar to typical human muscular dystrophy [36,37]. Intriguingly, the MA-5 treatment significantly improved the expanded FWHM (Figure 2C). This suggests that in the DMD model, the administration of MA-5 smoothed the movement of muscle contraction and relaxation cycle and increased the thrash rate.

For the *dys-1* (*eg33*) mutants, a severely fragmented mitochondrial network shown in the BWM cells of young adults, was improved by prednisone and NaGYY treatments, and restored to motility [31,38]. We, therefore, tested the effect of MA-5 on improvements in mitochondrial structure. As shown in Figure 3A,B, the fragmented network (1.74 ± 0.08 μm) was significantly improved by MA-5 treatment (2.76 ± 0.12 μm). Furthermore, it was confirmed that the accumulation of Ca^2+^ in the mitochondria ([Ca^2+^]_mito_), due to the *dys-1* defect, also improved with MA-5 treatment (Figure 3A,C). These results indicated that MA-5 could improve the health of *C. elegans* DMD model.

### 2.3. MA-5 Ameliorates Muscular Mitochondrial Perturbations with Rotenone Treatment

To evaluate the effect of MA-5 on mitochondrial perturbations in BWM cells, WT adults synchronized on day 1 (D1) were treated with a low concentration of 2 μM rotenone, an inhibitor of ETC complex I. The fluorescent signals of the mitochondria-specific ROS generation with MitoTracker Red CMXRos significantly increased after exposure to rotenone for 6 h (Figure 4). In contrast, the addition of MA-5 at the same time as rotenone exposure significantly reduced the subsequent ROS production from the mitochondria of the BWM cells (Figure 4).

Continuous treatment of the SD1347 wild-type strain carrying the *ccIs4251* transgene, mtGFP and nuclear targeted GFP (nucGFP), with 2 μM rotenone from synchronous D1 adulthood for 48 h significantly suppressed body growth associated with increased muscle mass (Figure 5A). Using confocal fluorescent microscopy, mitochondrial network fragmentation and nuclear destruction were also observed in BWM cells (Figure 5C–E). However, the administration of MA-5 significantly improved not only body growth retardation but also mitochondrial and nuclear damage (Figure 5A,C–E). In addition, rotenone reduced the total amount of endogenous ATP to one-third of WT, whereas MA-5 treatment was able to rescue it to two-thirds of WT (Figure 5B).

### 2.4. Alleviation of PD Model Progression by MA-5

Low concentrations of rotenone and 6-OHDA are known to be the PD stressors in mammals [21,22,23,24]. Rotenone and 6-OHDA have also been reported to induce *C. elegans* dopaminergic neurodegeneration visualized in the TG2435 strain carrying *dat-1p::GFP* as PD models [18,19,20]. Here, we confirmed the appearance of GFP fluorescence puncta in dopaminergic CEP neurons 24 and 48 h after treatment of synchronized L4 larvae with 2 μM rotenone (Figure 6). The number of puncta increased with the exposure time, indicating that 2 μM rotenone exposure caused the progression of dopaminergic neurodegeneration in *C. elegans* as a PD model (Figure 6B). On the other hand, rotenone-induced neurodegeneration was significantly suppressed by the administration of MA-5, suggesting that MA-5 alleviates not only muscle damage but also neuronal dysfunction that arises with chronic mitochondrial perturbation (Figure 6A,B).

### 2.5. Suppression of Mitochondrial Swelling in C. elegans Mitofilin immt-1 Mutant with MA-5

*C. elegans*. has two homologs of mitofilin, IMMT-1 and IMMT-2. These are not redundant, as each mutation affects the mitochondrial morphologies of the same cell and the effects of double mutations are additive [39,40]. In particular, *immt-1* defects cause severe local swelling of the mitochondria by disrupting the cristae morphology. In mammalian cells, MA-5 interacts with mitofilin and modulates the mitochondrial inner membrane organizing system (MINOS) [5]. We, therefore, investigated whether MA-5 could suppress the mitochondrial swelling caused by *immt-1* mutant in *C. elegans*. Muscle mitochondria visualized with the *ccIs4251* transgene mtGFP showed that irregularly enlarged mitochondria were increased in the *immt-1* mutant compared to the wild type (Figure 7A,B). When the *immt-1* mutants were cultured on NGM-plates containing MA-5 from L4 larvae to adulthood, the number of enlarged mitochondria were significantly reduced (Figure 7A,B). In addition, transmission electron microscopy showed that not only was the mitochondrial diameter suppressed but the deformation of the cristae morphology in the *immt-1* mutant was also suppressed by MA-5 treatment (Figure 7C). These results suggest that MA-5 functions stabilize and repair MINOS in *C. elegans*. muscles.

## 3. Discussion

Human DMD is an X-linked muscle-wasting disease caused by the loss of dystrophin protein, a rod-shaped cytoskeletal protein that is primarily expressed in muscles. In the absence of dystrophin, the structural integrity of the sarcolemma is lost and leads to disruption in skeletal muscle signaling, such as nitric oxide, ROS production pathways and Ca^2+^ cycles [41]. In particular, Ca^2+^ cycling between the sarcoplasmic reticulum (SR) and the cytoplasm is essential for the normal muscle contraction and relaxation cycle. It is also reported that in cardiomyocytes from *mdx* mice, an animal model of DMD, elevated [Ca^2+^]_mito_ is associated with the excessive opening of the mitochondrial permeability transition pore (mPTP), loss of mitochondrial membrane potential and mitochondrial depolarization [42]. It leads to substantial structural damage to the mitochondria and ultimately promotes cell death. Similar to human muscular dystrophy, an abnormal increase in [Ca^2+^]_cyto_ is observed in the *C. elegans**. dys-1* mutants, which in turn activates various matrix metalloproteinase-mediated extracellular matrix degradation [30,43]. First of all, the fluorescent MA-5, BODIPY-MA-5, was used to evaluate the efficiency and predominant penetration into the mitochondria of the *C. elegans*. tissues (muscle, germline, intestine and mammalian cells) [5] (Figure 1). Here, we also confirmed a muscle tone disorder in which [Ca^2+^]_cyto_ accumulates longer in the *dys-1 (eg33)* mutants immobilized with microspheres (Figure 2, Appendix A). In addition, the over-accumulation of [Ca^2+^]_mito_ levels in BWM cells of the *dys-1* mutant was observed with mitochondrial fragmentation (Figure 3).

Intriguingly, we found that the mitochondrial homing drug MA-5 significantly improved elevated [Ca^2+^]_mito_ and mitochondrial fragmentation in the BWM cells of *dys-1* mutants (Figure 3). In addition, prolonged [Ca^2+^]_cyto_ cycles and decreased thrash movements were recovered by the administration of MA-5 (Figure 2). These suggest that MA-5 is a potential therapeutic agent that works by ameliorating mitochondrial dysfunction in DMD, as demonstrated using the DMD model of *C. elegans*. Ellwood et al. [31] have identified that the use of hydrogen sulfide (H_2_S) supplementation (GY4137 and AP39) acts similar to prednisone and improves neuromuscular health using the *C. elegans* DMD model. They also find a decline in total sulfide and H_2_S-producing enzymes in dystrophin/utrophin knockout mice, suggesting the deficit with H_2_S may contribute to DMD pathology. On the other hand, the loss of Ca^2+^ homeostasis in *C. elegans* DMD model does not appear to be corrected by either prednisone or H_2_S supplementation [31]. Therefore, MA-5 may act by a different mechanism than H_2_S supplementation and prednisone. Interestingly, pharmacological activation of Sarco/endoplasmic reticulum Ca^2+^-ATPase (SERCA) with CDN1163 ameliorates dystrophic phenotypes in *m*dx mice [44]. The administration of CDN1163 reduced [Ca^2+^]_cyto_ levels in vitro and ex vivo, reversed the mitochondrial swelling, increased OCR and reduced ROS production in isolated mitochondria of *mdx* mice. Taking together, controlling Ca^2+^ homeostasis in muscular mitochondria and cytoplasm by MA-5 and CND1163 is effective in treating and alleviating muscular dystrophy.

The mitochondria generate ROS as an intrinsic by-product of ATP synthesis. The generation of ATP and ROS in healthy mitochondria is generally coupled [45,46]. On the other hand, mitochondrial dysfunction causes two detrimental consequences, decreased ATP synthesis and increased ROS production. To artificially cause such disorders, low concentrations of rotenone chronic exposure were applied in this study. MA-5 significantly suppressed the increase in mitochondrial ROS and the decrease in endogenous ATP levels (Figure 4 and Figure 5B). In addition, rotenone-induced the fragmentation of the mitochondrial network and the nuclear destruction of BWM cells was suppressed (Figure 5C–E). Moreover, chronic exposure of rotenone causes dopaminergic neurodegeneration in *C. elegans* as a PD model [19,20]. MA-5 alleviated dopaminergic neurodegeneration in the rotenone-treated PD model (Figure 6). In cultured mammalian neuronal cells, a recent mechanism has reported that rotenone (as one of the PD stressors) promotes the translocation of Parkin to mitochondria and increases the interaction between Parkin and mitofilin [24]. It finally causes ubiquitination-induced mitofilin degradation. Previously, we have shown that MA-5 binds directly to mitofilin and stabilizes the cristae structure. This promotes the oligomerization of ATP synthase and supercomplex formation, thereby increasing local ATP production as a mitochondrial-homing activity [6]. Thus, this pharmacological effect of MA-5 may be widely conserved in metazoans, from nematodes to humans. MA-5 may bind to another *C. elegans* mitofilin, IMMT-2 molecules, because the *immt-1*-deficient mutation also improved mitochondrial hypertrophy and cristae deformity (Figure 7). Taken together, MA-5 mitochondrial homing activity increases ATP production and reduces ROS levels, and MA-5 mitofilin binding may suppress the ubiquitination or degradation of mitofilin by Parkin even in the presence of PD stressors. Further work on the verification of this working hypothesis is needed in the future.

In conclusion, it is becoming increasingly clear that mitochondrial dysfunction plays a causal role in a number of neuromuscular diseases including DMD, PD, mitochondrial myopathy, amyotrophic lateral sclerosis and Alzheimer’s disease. This work demonstrates the beneficial effect of MA-5 on *C. elegans* DMD and PD models. We also found that the mitochondrial homing drug MA-5 significantly improves mitochondria Ca^2+^ homeostasis, in addition to the previously known ATP production and ROS reduction. MA-5 may act by a unique mechanism through its interaction with mitofilin and may be more beneficial when used in combination with other agents, such as H_2_S donors and prednisone.

## 4. Materials and Methods

### 4.1. C. elegans Strains and Culture Conditions

The standard procedures for *C. elegans* maintenance were followed using nematode growth media (NGM) agar plates with Escherichia coli OP50 as a food source and incubator at 20 °C [47]. MA-5 (Hayashi K-I, Okayama University of Science) and the ETC inhibitor rotenone (Millipore Sigma, Burlington, MA, USA) were applied to plates at a final concentration of 10 and 2 μM, respectively. Worms were age-synchronized from the eggs and allowed to grow to the designated day. The strains used in this study are as follows: wild-type N2, TG2435: *vtIs1* [*dat-1p::GFP + rol-6(su1006)*] V, ATU2301: *goeIs3* [*myo-3p::SL1::GCamP3.35::SL2::unc54 3’UTR + unc-119(+)*] V, *acels1*[*myo-3p::mitochondrial LAR-GECO+myo-2p::RFP*] II, ATU2305: *dys-1(eg33) goeIs3* [*Pmyo3::GCaMP3.35::unc-54-3’utr, unc-119(+)*], *aceIs1*[*Pmyo3::mitochondrial LAR-GECO + Pmyo2::RFP*] [34], ATU3301: *ccIs4251* [*(pSAK2) myo-3p::GFP::LacZ::NLS + (pSAK4)myo-3p::mitochondrialGFP+dpy-20(+)*] I, *acels1* II, ATU3305::*dys-1(eg33) ccIs4251* [*Pmyo3::nucGFP-LacZ + Pmyo-3::mitochondrial GFP*], *aceIs1* [*Pmyo-3::mitochondrial LAR-GECO+ Pmyo2::RFP*], SD1347: *ccIs4251* [*(pSAK2) myo-3p::GFP::LacZ::NLS+ (pSAK4) myo-3p::mitochondrial GFP + dpy-20(+)*] I and ATU3307: *ccIs4251* I, *acels1* II, *immt-1 (tm1730)* X.

### 4.2. BODIPY-Based Fluorescent-Conjugated MA-5 Contents in Mitochondria Assay

AIE™ Mitochondria Blue (AIEgen Biotech Co., Limited, Hong Kong), which stains mitochondrial with blue fluorescence at final concentration 25 μM was performed with 1 μM BODIPY-MA-5 [5] or 1 μM BODIPY (Fisher Scientific, Schwerte, Germany) for 30 min in adult wild-type (N2) worms on day 1. M9 buffer was used for washing *C. elegans*. The Z-stack images of BODIPY-MA-5, BODIPY, and AIE™ Mitochondria Blue fluorescence were presented using confocal microscopy. Fluorescent excitation and emission wavelengths were under 490/504, 490/504, 359/461, respectively.

### 4.3. Measurement of Body Length

The worms were synchronized, and their movement was recorded using stereomicroscopy (SMZ18; Nikon, Tokyo, Japan), a device camera (DP74; Olympus, Tokyo, Japan) and an imaging software (cellSens Standard 2.2; Olympus, Tokyo, Japan). Body length was measured from images captured by the software.

### 4.4. Thrashing Speed

To determine the body bending of the worms in the liquid, the thrashing speed of synchronized adult worms was measured in 1 mL of M9 buffer for 30 s. In total, 10 worms were measured for each treatment.

### 4.5. Microscopic Imaging

*C. elegans* BWM cells and their mitochondrial images were obtained using confocal laser-scanning microscopy (FluoView Olympus FV10i; Olympus, Tokyo, Japan). For observation, synchronized worms were washed with M9 buffer and mounted on a microscope slide (6.5 mm square, 20 μm deep well made with a water-repellent coating (Matsunami Glass Ind., Ltd., Osaka, Japan)) with 100 mM NaN_3_ solution. Muscular mitochondrial volume and length of mitochondrial networks were analyzed by Image J software (National Institutes of Health, Bethesda, MD, USA).

For the live imaging of the cytoplasmic Ca^2+^ cycling in BWM cells using GCaMP fluorescence (*goeIs3* transgene), the synchronized worms were washed and mounted with 2.5% polystyrene microspheres (0.10 μm, Polysciences Inc. Warrington, PA, USA). Time-lapse confocal images of cytosolic GCaMP fluorescence were acquired at room temperature (20~22 °C) by FV10i.

To examine BWM mitochondrial structures by transmission electron microscopy (TEM), four-day-old adult hermaphrodites of wild-type and *immt-1* mutants treated with or without MA-5 were used. Worms were fixed in 2% paraformaldehyde and 2.5% glutaraldehyde in 0.1 M cacodylate buffer (pH 7.4) at 4 °C, then washed 2 times in 0.1 M cacodylate buffer for 15 min each and fixed in 1% osmium tetroxide on ice for 90 min. They were dehydrated through a 50–95% ethanol series for 10 min each and 100% ethanol for 20 min three times, rinsed twice for 10 min in propylene oxide and embedded in resin containing TAAB Epon 812 (TAAB, Reading, UK) for 48 h at 60 °C. Ultra-thin sections (70 nm) cut with a diamond knife on an ultramicrotome (Leica Microsystems, Wetzlar, Germany) were mounted on copper grids, stained with 2% uranyl acetate at room temperature for 15 min, secondary-stained with lead stain solution (Sigma-Aldrich, Burlington, MA, United States) at room temperature for 3 min and then examined by TEM (H-7600, Hitachi, Ltd., Tokyo, Japan).

### 4.6. Measurement of Mitochondrial Ca^2+^ Levels

Mitochondrial Ca^2+^ and cytoplasmic Ca^2+^ in BWM cells were, respectively, observed by measuring the expression of the transgenes *aceIs1* and *goeIs3* as described recently from our group [34]. The Ca^2+^ concentration in muscle mitochondria ([Ca^2+^]_mito_) was calculated using the following equation [48].
[Ca^2+^]_mito_ = Kd·(R − Rmin)/(Rmax − R)
where Kd (12 μM) indicates the dissociation constant between Ca^2+^ and the LAR-GECO probe [49] and R indicates the ratio of fluorescence intensity of mtGECO to that of mtGFP of *ccIs4251* transgene in a constant area.

### 4.7. Dopamine Neuron Degeneration Measurement

Age-synchronized two-day-old adult worms with *dat-1p::GFP* (TG2435) were used in this experiment. Approximately 12 worms were analyzed for each condition. Images were obtained using confocal microscopy and ImageJ software was used to calculate the number of beads in all four cephalic (CEP) neurons [50].

### 4.8. Mitochondrial ROS Measurement

Wild-type (N2) one-day-old adult nematodes were incubated with rotenone for 6 h in the presence or absence of MA-5. Subsequently, worms were treated with 0.5 µM MitoTracker^®^ Red CMXRos (Fisher Scientific, Schwerte, Germany) and 25 μM AIE™ Mitochondria Blue (AIEgen Biotech Co., Limited, Hong Kong, China) for 3 h. Nematodes were then transferred to fresh NGM agar plates and incubated overnight. Images were obtained using confocal microscopy and fluorescence intensity was analyzed using FV10-ASW Viewer software (Olympus, Tokyo, Japan).

### 4.9. ATP Detection

Wild-type (N2) worms on two-day-old adults were collected in 100 µM M9 buffer for further ATP assays. An ATP determination kit (Molecular Probes, Eugene, OR, USA) was used to measure endogenous ATP levels, as previously reported for rotenone treatment [51].

### 4.10. Analysis of mitofilin/immt-1gene Mutation Mitochondrial Morphology

The synchronized *immt-1* mutant (ATU3307) on a four-day-old adult was used in this experiment. Approximately 50 muscle cell images were taken from 10 worms for each treatment. Images of mitochondrial morphology were observed using confocal microscopy and analyzed using ImageJ software.

### 4.11. Statistical Analysis

The one-way ANOVA with post hoc Tukey’s HSD and Dunn’s test were used for comparisons between groups as appropriate (R or Origin software). All data points including outliers were used for means and statistical significance. A *p* value of < 0.05 was considered significant. Different letters indicate significant differences between the groups.

## Figures and Tables

**Figure 1 ijms-23-09572-f001:**
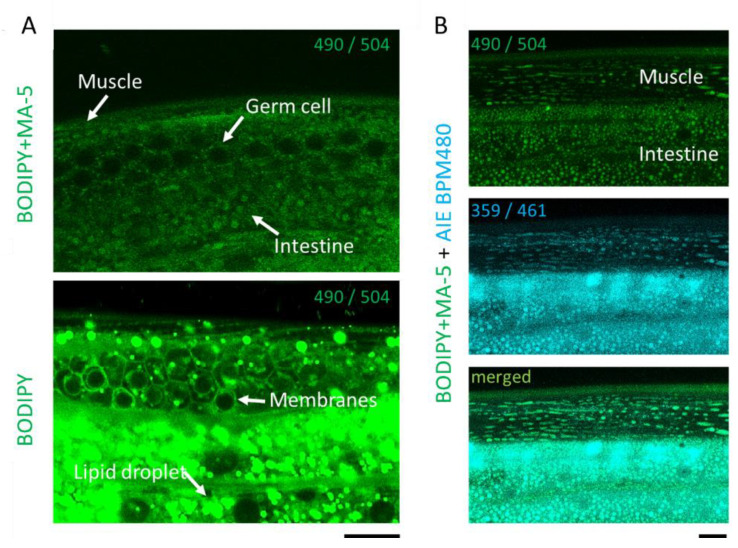
Penetration and homing activity of MA-5 into intact *C. elegans* mitochondria. (**A**) Differences between BODIPY-MA-5 and BODIPY. Fluorescent signals of BODIPY-MA-5 on mitochondria are indicated in muscle, germ cell and intestine (white arrows). BODIPY staining signals show in intestinal lipid droplets and germ cell membranes. (**B**) The z-stack images of body wall muscle cells of wild-type (N2) adults on day 2 were monitored by confocal microscopy. Localization of MA-5-BODIPY in mitochondria (indicated as green), Mitochondrial Maker AIE mitochondria blue (indicated as blue) and merged image demonstrating mitochondrial localization (indicated as cyan). Fluorescent excitation and emission wavelengths (nm) are indicated as numbers in each picture excitation⁄ emission (nm). Scale bars represent 10 µm.

**Figure 2 ijms-23-09572-f002:**
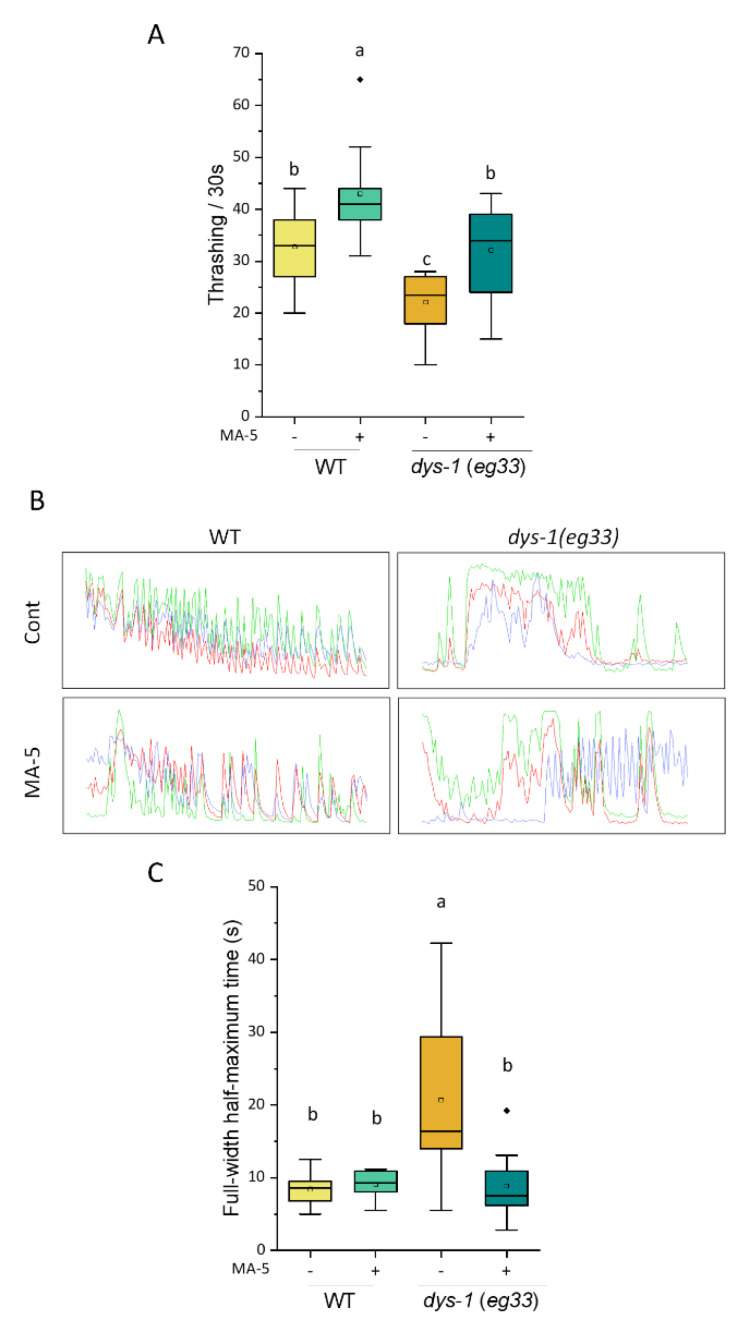
MA-5 suppressed DMD disease model symptoms on motor activity reduction and extension of cytoplasmic Ca^2+^ oscillations with muscle contraction. (**A**) Movement capacity of ATU3305 worms in liquid media. Locomotory performances were determined in 1ml M9 for each 30 s (*n* = 10 worms/treatment). Different letters indicate significant differences (*p* ≤ 0.05) by one-way ANOVA and Tukey’s HSD test with all data points including outliers. Transgenic *C. elegans dys-1* null mutation ATU3305 (*dys-1(eg33) ccIs4251* [*Pmyo3::nucGFP-LacZ + Pmyo-3::mitochondrial GFP*] *aceIs1* [*Pmyo-3::mitochondrial LAR-GECO+ Pmyo2::RFP*]) and wild-type ATU3301 (*ccIs4251* [*(pSAK2) myo-3p::GFP::LacZ::NLS**+(pSAK4)myo-3p::mitochondrialGFP + dpy-20(+)*] I, *acels1* II) were monitored on day 2 of adulthood. (**B**) Cytoplasmic Ca^2+^ oscillations with muscle contraction in different three muscle cells (indicated as green, red and blue) with wild-type (ATU2301) and *dys-1* null mutant (ATU2305) in the presence or absence of MA-5 treatment for 300 s. Transgenic *C. elegans dys-1* null mutation ATU2305 (*goeIs3* V, *aceIs1* II, *dys-1* (*eg33*) I) and wild-type ATU2301 (*goeIs3; aceIs1*) were monitored on day 2 of adulthood. (**C**) Peak width measured as the full width at half maximum (FWHM) of cytoplasmic Ca^2+^ was analyzed (*n* = 12–25/treatment). Different letters indicate significant differences (*p* ≤ 0.05) by Dunn’s test with all data points including outliers. Data are shown as box plots to indicate median (central line) and mean (square mark). Cont: control treated with 0.1% DMSO; MA-5: 10 μM; WT: wild type (strain ATU3301); Diamond mark: outlier data.

**Figure 3 ijms-23-09572-f003:**
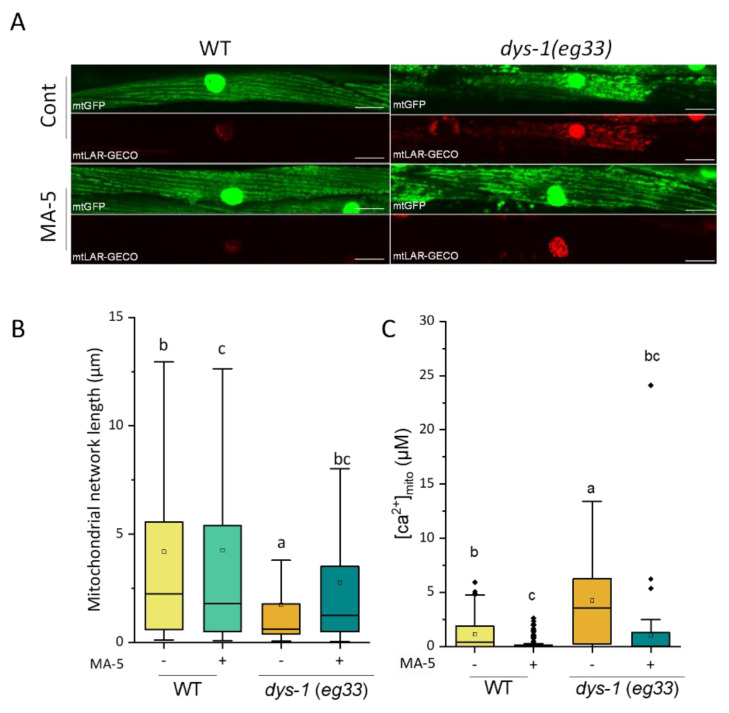
MA-5 suppressed DMD disease model symptoms on mitochondrial fragmentation and mitochondrial Ca^2+^ levels increase. (**A**) Representative images of the mitochondrial morphologies (indicated as green), mitochondrial calcium signal (indicated as red) and merged observed. (Scale bar: 20 µm). Transgenic *C. elegans dys-1(eg33)* null mutant ATU3305 expressing mitochondria-targeted GFP (mtGFP) and mitochondrial calcium-targeted LAR-GECO (mtLAR-GECO) in body wall muscle cells were monitored on day 2 of adulthood. (**B**) The mitochondria network length in each muscle cell (*n* ≥ 900 mitochondria from 5–7 worms/treatment) treated with or without MA-5 of 2-day-old ATU3305 adults. Different letters indicate significant differences (*p* ≤ 0.05) by Dunn’s test. (**C**) Concentration of mitochondrial calcium in worms (*n* ≥ 60 from 6–12 worms/treatment). Different letters indicate significant differences (*p* ≤ 0.05) by Dunn’s test with all data points including outliers. Data are shown as box plots to indicate median (central line) and mean (square mark). Cont: control treated with 0.1% DMSO; MA-5: 10 μM; WT: wild type (strain ATU3301); Diamond mark: outlier data.

**Figure 4 ijms-23-09572-f004:**
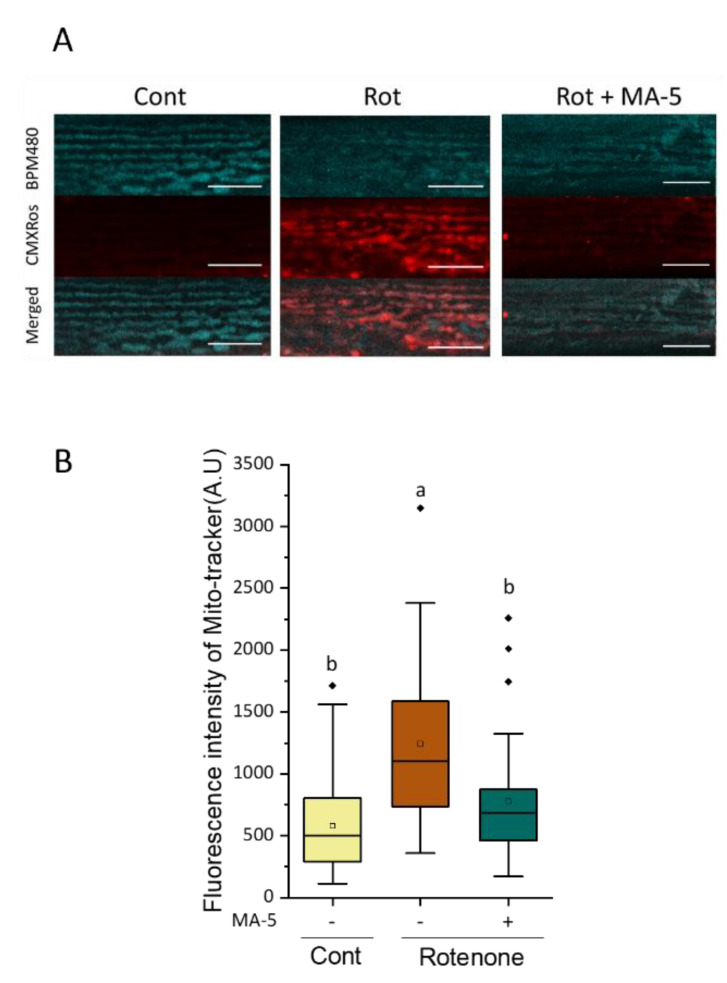
MA-5 suppressed rotenone-induced reactive oxygen species (ROS) levels after 6 h of treatment. Wild-type (N2) adult worms treated with rotenone applied with or without MA-5 on day 2. (**A**) Representative images of mitochondria stained by AIE™ Mitochondria Blue (indicated as blue) and MitoTracker^®^ Red CMXRos (indicated as red) and merged (scale bar: 10 µm). MitoTracker^®^ Red CMXRos was applied to evaluate the mitochondrial site of ROS generation and AIE™ Mitochondria Blue to determine the mitochondrial location in body muscle cells. (**B**) The intensity of MitoTracker^®^ Red CMXRos fluorescence as an indicator of ROS production (*n* ≥ 30 mitochondria from 6–14 worms/treatment). Data are shown as box plots to indicate median (central line) and mean (square mark). Different letters indicate significant differences (*p* ≤ 0.05) using Dunn’s test with all data points including outliers. Cont: control treated with 0.6% of DMSO; MA-5: 10 μM; Rot: 2 μM Rotenone; Mitoblue: AIE™ Mitochondria Blue; CMXRos: MitoTracker^®^ Red CMXRos; Diamond mark: outlier data.

**Figure 5 ijms-23-09572-f005:**
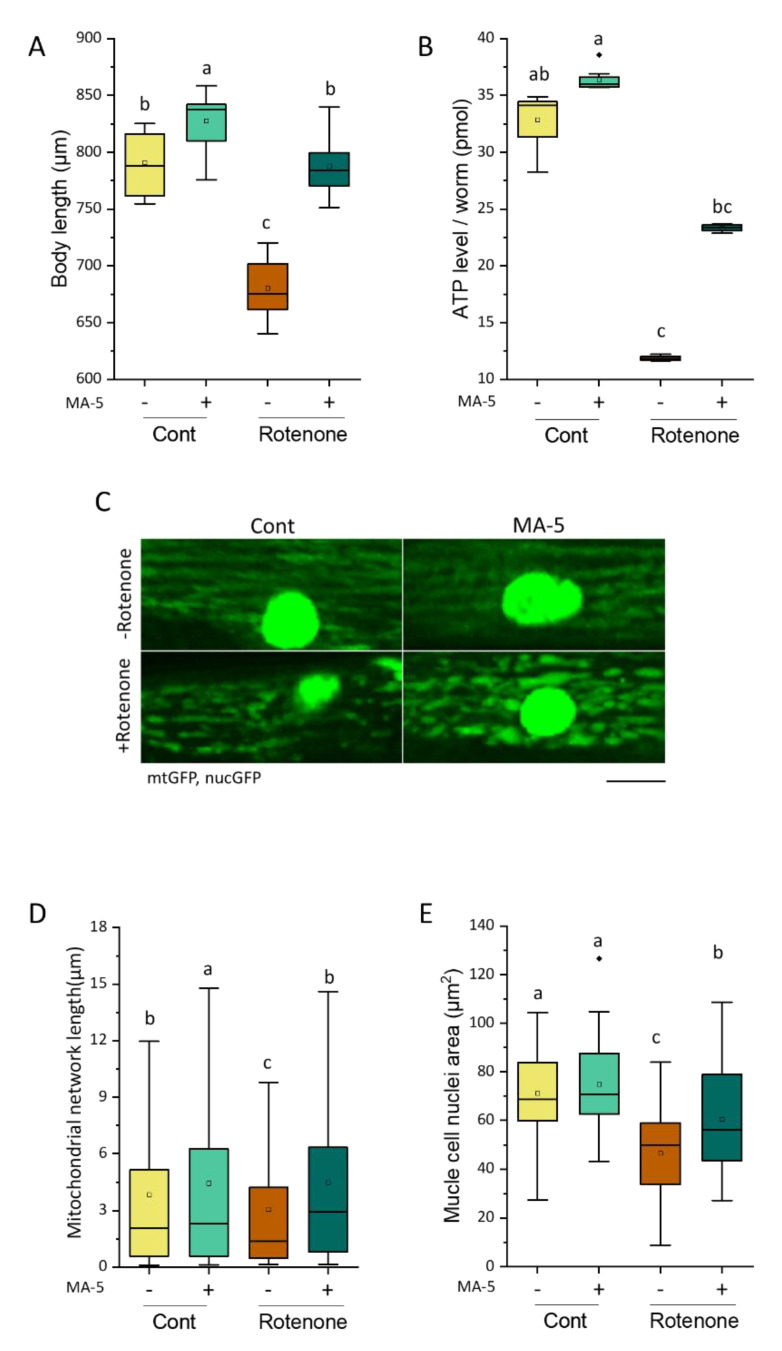
MA-5 suppressed rotenone-induced growth retardation, ATP level decline, mitochondrial fragmentation and nuclear breakdown. (**A**) The body length of N2 worms following rotenone treatment with or without MA-5 (*n* = 11–20 worms/treatment) was monitored after 48 h. Different letters indicate significant differences (*p* ≤ 0.05) by one-way ANOVA and Tukey’s HSD test. (**B**) ATP levels in N2 worms following rotenone treatment with or without MA-5 (*n* = 12–18 worms/treatment) were monitored after 48 h. ATP levels were assessed using an ATP determination kit (Molecular Probes, Eugene, OR, USA). Different letters indicate significant differences (*p* ≤ 0.05) using the Dunn’s test with all data points including outliers. (**C**) Representative images of mitochondrial and nuclear morphology after 48 h treatment (scale bar: 8 μm). Synchronized SD1347 worms (*ccIs4251 [(pSAK2) myo-3p::GFP::LacZ::NLS+ (pSAK4) myo-3p::mitochondrial GFP + dpy-20(+)*]) I following rotenone treatment with or without MA-5 were monitored after 48 h. (**D**) The mitochondrial network length (*n* ≥ 600 mitochondria from 5–8 worms/treatment) and (**E**) muscle cell nuclei area in SD1347 worms following rotenone treatment with or without MA-5 (*n* ≥ 20 nuclei from 6–8 worms/treatment) were monitored after 48 h. Different letters indicate significant differences (*p* ≤ 0.05) by one-way ANOVA and Tukey’s HSD test with all data points including outliers. Data are shown as box plots to indicate median (central line) and mean (square mark). Cont: control treated with 0.1% of DMSO; MA-5: 10 μM; Rotenone: 2 μM; Diamond mark: outlier data.

**Figure 6 ijms-23-09572-f006:**
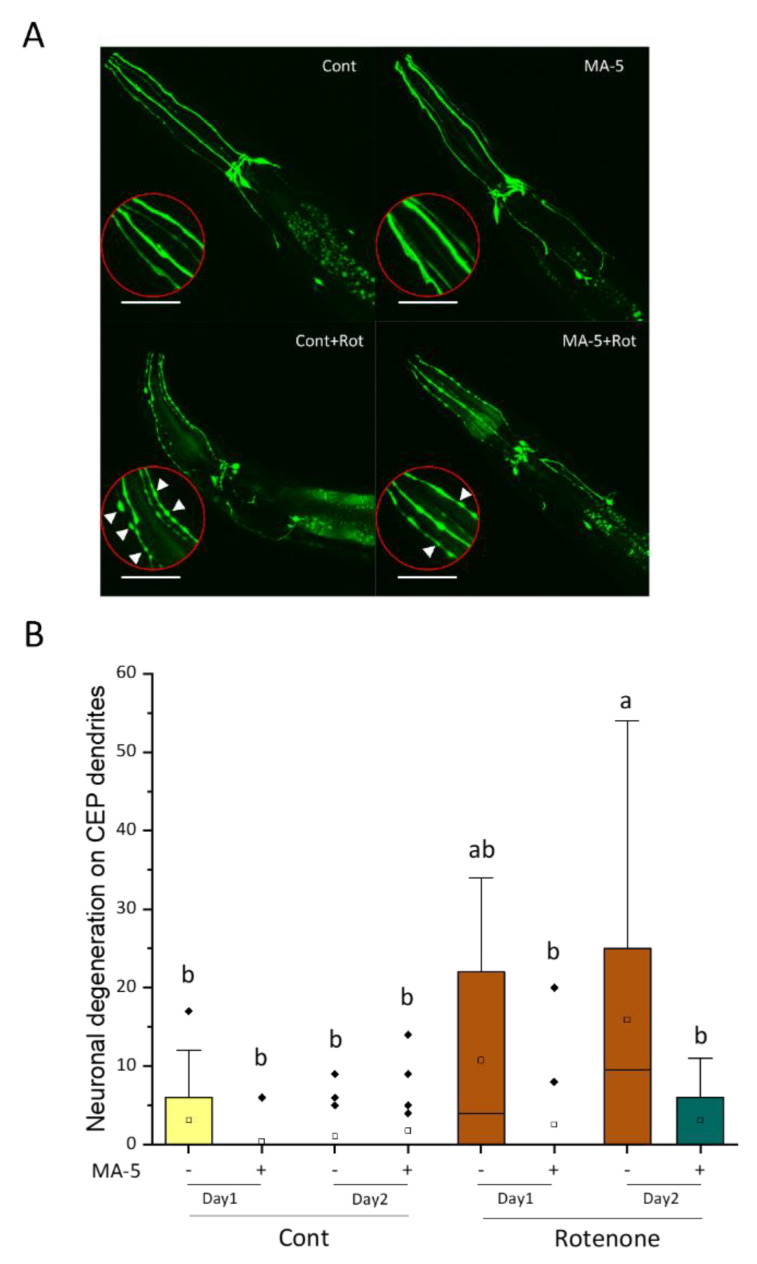
MA-5 suppresses rotenone-induced neurodegeneration in dopaminergic cephalic (CEP) neurons. Transgenic *C. elegans*. TG2435 (*vtIs1* [*dat-1p::GFP + rol-6(su1006)*] V) expressing dopaminergic neurons tagged with a green fluorescent protein (GFP) following rotenone treatment with or without MA-5 were monitored after 24 and 48 h. (**A**) Representative images of dopamine neuron degeneration under different treatments for 48 h. (Scale bar: 50 µm). White arrowheads in the enlarged red circles indicate neuronal processes that exhibit abnormally discontinuous GFP signals. (**B**) Frequency of the four CEP blebs along the dendrites in animals (*n* = 18–20/treatment). Data are shown as box plots to indicate median (central line) and mean (square mark). Different letters indicate significant differences (*p* ≤ 0.05) using one-way ANOVA and Tukey’s HSD test with all data points including outliers. Cont: control treated with 0.1% of DMSO; MA-5: 10 μM; Rot: 2 μM rotenone; Diamond mark: outlier data.

**Figure 7 ijms-23-09572-f007:**
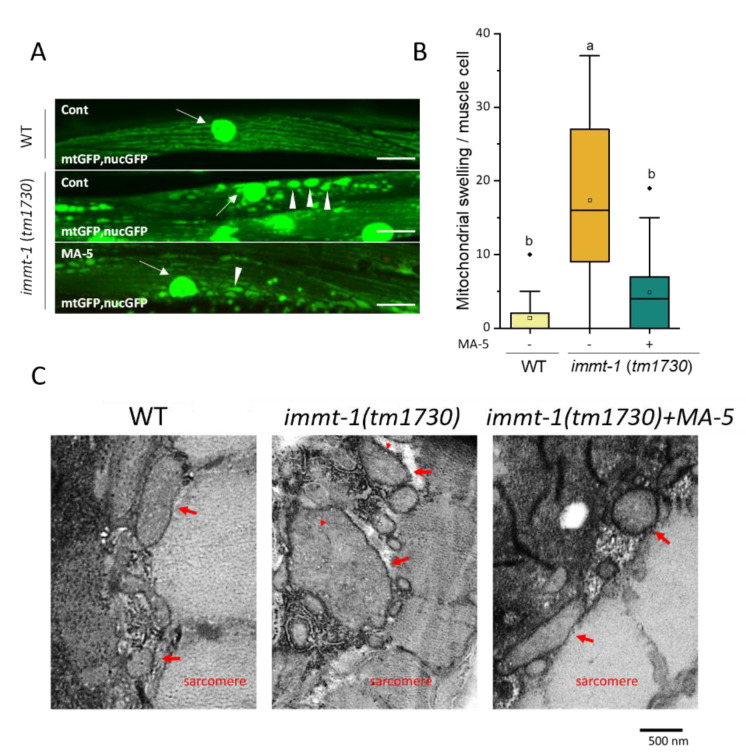
MA-5-induced suppression in mitofilin/*immt-1* gene mutation against abnormal mitochondrial morphology. *immt-1 (tm1730)* null mutation ATU3307 worms expressing mitochondria-targeted green fluorescent protein (mtGFP) and nuclear-targeted GFP–LacZ (nucGFP) in body wall muscle cells were treated with or without MA-5 from L4 to adult stage on day 4. (**A**) Representative images of mitochondrial morphologies are shown (scale bar: 20 µm). White arrows indicate body wall muscle cell nuclei, and white arrowheads indicate abnormally swollen mitochondria. (**B**) The number of abnormal mitochondria in each muscle cell (*n* ≥ 35 from 6–8 worms/treatment). Data are shown as box plots to indicate median (central line) and mean (square mark). Different letters indicate significant differences (*p* ≤ 0.05) using Dunn’s test with all data points including outliers. (**C**) Observation of the mitochondria using transmission electron microscopy (scale bar: 500 nm). Red arrows indicate abnormal mitochondria and red arrowheads indicate abnormal cristae. WT: wild-type strain ATU3301 as mentioned before; Cont: control treated with 0.1% DMSO; MA-5: 10 μM; Diamond mark: outlier data.

**Table 1 ijms-23-09572-t001:** Median lifespan of ATU3301 animals ^a^ treated with MA-5.

MA-5 Final Concentraation	Median Lifespan (Days) ^b^
control (0 μM)	13.6 ± 0.6
5 μM	15.3 ± 0.5
10 μM	14.8 ± 0.4
20 μM	14.8 ± 0.3

^a^ ATU3301 (ccIs4251 [(pSAK2) myo-3p::GFP::LacZ::NLS + (pSAK4) myo-3p::mitochondrial GFP + dpy-20(+)] I, acels1 [myo-3p::mitochondrial LAR-GECO + myo-2p::RFP]II) in N2 wild-type background [34]. ^b^ Median life expectancy is the age at which half of the population died. One test plot was approximately *n* = 50, and each experiment was performed in triplicate. There is no significant difference between control and every MA-5 treatment.

## Data Availability

Not applicable.

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
