# Peer review of "Mitochonic Acid 5 Improves Duchenne Muscular Dystrophy and Parkinson’s Disease Model of Caenorhabditis elegans"

_ijms, 2022, doi:10.3390/ijms23179572_

Round 1
Reviewer 1 Report
· The article "Mitochonic acid-5 improves Duchenne muscular dystrophy and 2 Parkinson’s disease model of Caenorhabditis elegans" gives an interesting insight into the mechanistic aspects of the drug MA-5 in DMD and PD model in C.elegans. A few queries, if addressed, could improve its overall quality.
What is the toxicity profile of the drug MA-5 in C. elegans?
· How reproducible and translatable are the results found in C. elegans models in higher order animals and humans in general?
· How is the word “progression” in the result 2.4 (Alleviation of PD model progression by MA-5) justified? Measuring dopamine signal intensity alone cannot be a measure of disease progression. Additionally, there is no experiment using different time points in C. elegans model that could represent actual stage specific progression in PD.
Author Response
Dear Reviewer,
Thank you very much for your kind advices and corrections.
Please see the attachment.
Sincerely yours,
Atsushi

Reviewer 2 Report
Review of IJMS 1845640 MA-5 in DMD & PD
This paper studies the effect of mitochonic acid-5 (MA-5) on improving disturbed mitochondria and mitochondrial function in three C.Elegans models which have (i) DMD (dystrophin) gene mutation (eg33); (ii) mitochondrial dysfunction due to Mitofilin1 (immt-1) gene mutation (tm1730); (iii) PD (Parkinson’s Disease) model.
For the PD model, the authors study the capacity of MA-5 to partially reverse the mitochondrial effects of rotenone – [a quinone derivative of the plant hormone indole-3-acetic acid (IAA)] - in dopaminergic neurons of either wild-type or PD-modelled (? Rol-6 su1006) C.Elegans strains. For each parameter studied the authors find significant improvement with MA-5. This finding could be important as a first step in a pathway towards therapeutic research.
Apart from 2 major points (below), the paper is well-written, with only a few suggested text alterations needed to improve its clarity.
Major point
1. For the PD model, it is unfortunately not immediately clear whether this is wild-type C.elegans exposed to rotenone (giving dopaminergic neurone mitochondrial dysfunction), or whether the rotenone is applied to a specific dopaminergic-neurone-instability strain (? Rol-6 su1006) used as a genetic model of susceptibility for Parkinson’s disease. The authors need to be more explicit about this in the Introduction. Since most human Parkinson’s disease is in the elderly and has a presumed vascular or neurodegenerative basis with minimal genetic component, the authors also need to clarify (particularly if a genetically more-susceptible strain is being used) which is the human genetic type of PD (by age-onset and gene involved) that is modelled in C-elegans. If this is a young-onset and relatively rare form of PD, they should also comment on the relevance or not of their findings (and the therapeutic research pathways that may result from this) to the majority of human Parkinson’s disease cases.
2. Unfortunately in each of the sets of data presentation (and presumably in the calculation of means and statistical significance), the authors have excluded ‘outliers’ without giving any explanation of the rationale or criteria for doing so. Without that information it is not possible to assess the value of this paper, or the true significance of the results. This must be addressed in the Results (and explained in the M&M and Discussion sections). Means and statistical significance values should be given both without any exclusion of outliers, and with exclusions where the authors can show valid justification to do so. The justification should include an indication on the ‘box & whisker’ plots of all the data points, so that the ‘outliers’ can be viewed in context. This may require repeat ‘box & whisker’ or scatter plots as (perhaps) supplementary material.
Minor points
2. Abstract. Line 21 Typo. ‘modulates’ rather than ‘modulate’
3. Introduction. Lines 41-42. Currently this reads as if IAA is a derivative of plant hormone. Please change to : ‘was synthesised as a derivative of the plant hormone, indole-3-acetic acid (IAA).'
4. Intro. Lines 49-50. ‘…the DMD gene.’ Although ‘DMD gene’ is strictly correct terminology, I think it is usually helpful to expand this to ‘…the DMD (dystrophin-encoding) gene.’
5. Intro. Lines 52-53. ‘Except exercise, some mitochondrial medicines improve the capacity…’
What do the authors mean as the exception; is it 'except during exercise' ? This sentence needs re-writing. In doing so, it might help to specify : ‘…some mitochondrial-targeted medicines (such as quinones) improve…’
6. Intro. Line 55. ‘…the performance of such quinones…’ . Do the authors mean ‘…performance in-vivo…’ (ie. therapeutic effectiveness) ?; if so, please add that.
7. Intro. Lines 64-65. Separate sentences are required here. Do the authors mean ? :
These disease models.....develop novel drugs [16-19] in addition to enabling study of genetic variation. (new sentence) Chronic exposure to low concentration .... ?
Or, do the authors mean ?: ‘…develop novel drugs [16-19]. (new sentence) In addition to induced genetic variation being a cause of dopaminergic neurodegeneration in C.elegans, this can also be achieved by chronic exposure to low concentration.....' ? Please amend/re-write accordingly.
8. Intro. Line 68. ‘…alleviate the symptoms…. ? This might imply ‘all symptoms’. It would be helpful here to define which 'symptom' is being studied. Also 'Symptoms' suggests conscious experience of a problem. It might be better to say : ' could alleviate the manifestations in body wall muscle (BWM) cells in a C.Elegans DMD model; and alleviate the changes in dopaminergic neurons in a C.Elegans PD model.
9. Results 2.1. Line 76. ‘…highly merged…’ What do the authors mean here ? Do the authors mean 'fully infiltrated' or 'perfused'. Please add a note of explanation if ‘highly merged’ is the correct term.
10. Results 2.1. Line 78. ‘..BODIPY…’. Is this BODIPY alone (without MA-5) ? If so, please write ‘…BODIPY on its own…’ (or something similar).
11. Figure 1. Typo ? The vertical labelling reads as ‘BODYPI’ all 3 times. Should this be ‘BODIPY’ ?
12. Figure 1. Caption. Line 84-85. ‘BODIPY staining signals in intestinal lipid droplets and germ cell membranes.’ As a sentence, this needs a verb. eg. ‘…BODIPY staining signals show in intestinal...’
13. Results. 2.3. Line 153. ‘…improved…’ It is probably better here to say 'reduced' rather than 'improved. 'Improved' suggests an additional interpretation of the observation, rather than just the observation itself.
14. Results. 2.3 Line 155. ‘…Body growth retardation, including muscle mass development, was observed…’ As written, this could imply that muscle mass is increased; it is probably better to say: ‘…including of muscle mass development…’
15. Materials & Methods. 4.6. Line 380. ‘…described [39].’ Please rewrite as : ‘…as described recently from our group. [39].’
16. Materials & Methods. 4.9. Line 404. ‘…was collected…’ Should be ‘…were collected…’
Author Response
Dear Reviewer,
Thank you very much for your kind advices and corrections. We have corrected all points following your advices. The revised version will be more suitable for readers of the manuscript.
Please see the attachment.
Thanks again,
Sincerely yours,
Atsushi

Round 2
Reviewer 2 Report
Re-review of vs2 of IJMS 1845640
The authors have attended satisfactorily (apart from a couple of minor phrasing corrections as below) to the points raised in the first review. In particular they confirm that the ‘outliers’ in the box & whisker plots are included in the data analyses. However:….
Point A) I regret and apologise that I had not appreciated that ‘outliers’ are created automatically to a set formula in ‘Box & Whisker’ chart-drawing programmes. However, because that formula can be adjusted, the authors do need to re-check the definition of the upper and lower edges of the boxes and of the ends of the whiskers in all their Bar and Whisker charts, and state that definition in the Legend to Figure 2, and in ‘Methods - Statistical Analysis’, indicating that it applies to all such diagrams. For examples: see: https://chartio.com/learn/charts/box-plot-complete-guide/
At present the authors write (line 162-3; 175-6; 195-196; 225; 247-8; 275-6; ): ‘… Data are shown as box plots to indicate median, mean (square mark), and standard deviation. …’ If the box edges were ‘one standard deviation’, they would be symmetrically placed about the mean value (which they do not appear to be in several of the plots in the paper here). The box edges could be 32nd and 68th centile, which would coincide with 1 SD, but only for a normally-distributed dataset (which most of these appear not to be). The usual box plot design is for the lower and upper box edges to be at the 1st and 3rd quartile values respectively, (25th and 75th centile). The whiskers then usually represent the most extreme data point which is within 1.5x the inter-quartile range (IQR) from that quartile. However, the whiskers can also be set to be at reciprocal centile values (eg. 2nd and 98th centile). The authors do therefore need to re-define what their box edges and whiskers represent.
Point B) The authors also need to indicate how many data points contribute to each box plot (? 10-12 in each case ?)
Phrasing suggestions :
Lines 74-76: ‘…As an example of studying PD, administration of the neurotoxins 6-hydroxydopamine (6-OHDA) visualizes dopaminergic neurodegeneration in the dat-1p::GFP transgenic C. elegans TG2435 [18,19].’
Would this read better as ? : ‘…As an example, in studying PD, administration of the neurotoxin 6-hydroxydopamine (6-OHDA), leads to visualisation of dopaminergic neurodegeneration in the dat-1p::GFP transgenic C. elegans TG2435 [18,19].’
Line 82: ‘… the target protein can be replaced by the human gene [26,27].
A protein cannot be replaced by a gene. Would this read better as ? :
‘… the target protein can be replaced by one encoded by the human gene [26,27].
Author Response
Dear reviewer and editor,
Thank you very much for your kindness and helpful comments.
Point A) Sorry. We miswrote the SD in the legends of box plot data. The bars show the 25th and 75th centile. Therefore, all figure legends "Data are shown as box plots to indicate median, mean (square mark), and standard deviations. " were changed to "Data are shown as box plots to indicate median (central line) and mean (square mark). ". (lines: 161-162; 175; 195-196;225-226; 247-248 and 275-276).
Point B) Thank you. All sample numbers (n) were mentioned in the legend of each figure. (lines: 160; 171; 173; 194-195; 222; 223 and 275).
According to your phrasing suggestions, we changed the text lines 74-76 and 82.
Thank you again for your support.
Sincerely yours,
Atsushi